# Pronunciation-Lexicon Free Training for Phoneme-based Crosslingual ASR via Joint Stochastic Approximation

## Abstract

Recently, pre-trained models with phonetic supervision have demonstrated their advantages for crosslingual speech recognition in data efficiency and information sharing across languages. The Whistle approach relaxes the requirement of gold-standard human-validated phonetic transcripts and adopts weakly-phonetic supervision; however, a limitation is that a pronunciation lexicon is needed for such phoneme-based crosslingual speech recognition. In this study, we aim to eliminate the need for the pronunciation lexicon and propose a latent variable model based method, with phonemes being treated as discrete latent variables. The new method consists of a speech-to-phoneme (S2P) model and a phoneme-to-grapheme (P2G) model, and a grapheme-to-phoneme (G2P) model is introduced as an auxiliary inference model. To jointly train the three models, we utilize the joint stochastic approximation (JSA) algorithm, which is a stochastic extension of the EM (expectation-maximization) algorithm and has demonstrated superior performances particularly in estimating discrete latent variable models. Based on the Whistle multilingual pre-trained S2P model, crosslingual experiments on Polish (130h) and Indonesian (20h) are conducted. By using only 10 minutes of phoneme supervision, the new method, called as Whistle-JSA, performs close to crosslingual fine-tuning with the full set of phoneme supervision, and on par with the method of crosslingual fine-tuning with subword supervision. Furthermore, it is found that in language domain adaptation (i.e., utilizing cross-domain text-only data), Whistle-JSA outperforms the standard practice of language model fusion via the auxiliary support of the G2P model.

## 1 Introduction

In recent years, automatic speech recognition (ASR) systems based on deep neural networks (DNNs) have made significant strides, which benefit from large amounts of transcribed speech data. Remarkably, more than 7,000 languages are spoken worldwide (Ethnologue, 2019), and most of them are low-resourced languages. A pressing challenge for the speech community is to develop ASR systems for new, unsupported languages rapidly and cost-effectively. Crosslingual ASR have been explored as a promising solution to bridge this gap (Schultz & Waibel, 1998; Conneau et al., 2021; Babu et al., 2021; Zhu et al., 2021).

In crosslingual speech recognition, a pre-trained multilingual model is fine-tuned to recognize utterances from a new, target language, which is unseen in training the multilingual model. In this way, crosslingual speech recognition could achieve knowledge transfer from the pre-trained multilingual model to the target model, thereby reducing reliance on transcribed data and becoming one of the effective solutions for low-resource speech recognition. Most recent research on pre-training for cross-lingual ASR can be classified into three categories - supervised pre-training with graphemic transcription or phonetic transcription, and self-supervised pre-training. The pros and cons of the three categories have recently been discussed in (Yusuyin et al., 2024). Under a common experimental setup with respect to pre-training data size and neural architecture, it is further found in (Yusuyin et al., 2024) that when crosslingual fine-tuning data is more limited, phoneme-based supervised pre-training achieves the most competitive results and provides high data-efficiency. This makes sense since phonetic units such as described in International Phonetic Alphabet (IPA), are

exactly those sounds shared in human language throughout the world. In contrast, the methods using grapheme units face challenges in learning shared crosslingual representations due to a lack of shared graphemes among different languages.

A longstanding challenge in phoneme-based speech recognition is that phoneme labels are needed for each training utterance. Phoneme labels are usually obtained by using a manually-crafted pronunciation lexicon (PROLEXs), which maps every word in the vocabulary into a phoneme sequence. Grapheme-to-phoneme (G2P) tools have been developed to aid this process of labeling sentences from their graphemic transcription into phonemes, but such tools are again created based on PRO-LEXs. There are enduring efforts to compile PROLEXs and develop G2P tools (Novak et al., 2016; Mortensen et al., 2018; Hasegawa-Johnson et al., 2020) for different languages. Overall, the existing approaches of phoneme-based ASR heavily depend on expert labor and are not scalable to be applied to much more low-resource languages.

In this paper we are interested in reducing the reliance on PROLEXs in building phoneme-based crosslingual ASR systems, i.e., towards PROLEX free. In recognizing speech $x$ into text $y$, phonemes arise as intermediate states. So intuitively we propose to treat phonemes as hidden variables $h$, and construct a latent variable model (LVM) with pairs of speech and text $(x, y)$ as observed values. Basically, the whole model is a conditional generative model from Speech to Phonemes and then to Graphemes, which is referred to as a SPG model, denoted by $p_\theta(h, y|x)$. SPG consists a speech-to-phoneme (S2P) model $p_\theta(h|x)$ and a phoneme-to-grapheme (P2G) model $p_\theta(y|h)$, and is thus a two-stage model. Latent variable modeling enables us to train the SPG model, without the need to knowing $h$, by maximizing marginal likelihood $p_\theta(y|x)$. This is different from previous two-stage ASR model with phonemes as intermediate states, as reviewed later in Section 2. Learning latent-variable models usually involves introducing an auxiliary G2P model $q_\phi(h|y)$.

**Method contribution.** Note that phonemes take discrete values, and recently the joint stochastic approximation (JSA) algorithm (Xu & Ou, 2016; Ou & Song, 2020) has emerged for learning discrete latent variable models with impressive performance. In this paper we propose to apply JSA to learn the SPG model, which is called the SPG-JSA approach in general. Particularly, we develop Whistle-JSA, a specific instantiation of SPG-JSA, where the S2P model is initialized from a pre-trained phoneme-based multilingual S2P backbone, called Whistle (Yusuyin et al., 2024). In practice, when viewing phonemes as labels, we combine supervised learning over 10 minutes of transcribed speech with weak phoneme labels and unsupervised learning over a much larger dataset without phoneme labels. Bootstrapping from a good S2P backbone (like Whistle) and providing few-shots samples of latent variables (such as 10 minutes of weak phoneme labels) is found to be important to make SPG-JSA successfully work in the challenging task of crosslingual ASR.

**Experiment contribution.** Crosslingual experiments on Polish (130h) and Indonesian (20h) are conducted. By using only 10 minutes of phoneme supervision, Whistle-JSA obtains close performance to crosslingual fine-tuning with the full set of phoneme supervision, and is on par with the method of crosslingual fine-tuning with subword supervision. Furthermore, it is found that in language domain adaptation (i.e., utilizing cross-domain text-only data), Whistle-JSA outperforms the standard practice of language model fusion via the auxiliary support of the G2P model.

## 2 RELATED WORK

**Crosslingual ASR.** Multilingual and crosslingual speech recognition has been studied for a long time (Schultz & Waibel, 1998). Modern crosslingual speech recognition typically fine-tunes a multilingual model pre-trained on multiple languages. Most recent research on multilingual pre-training can be classified into three categories - supervised pre-training with graphemic transcription (Li et al., 2021; Pratap et al., 2020; Tjandra et al., 2023; Radford et al., 2023) or phonetic transcription (Li et al., 2020; Zhu et al., 2021; Tachbelie et al., 2022; Yusuyin et al., 2023), and self-supervised pre-training (Conneau et al., 2021; Babu et al., 2021; Pratap et al., 2024). It is shown in (Yusuyin et al., 2024) that when crosslingual fine-tuning data is more limited, phoneme-based supervised pre-training can achieve better results compared to subword-based supervised pre-training and self-supervised pre-training. However, phoneme-based crosslingual fine-tuning in (Yusuyin et al., 2024) requires phoneme labels for every training utterance from the target language, which relies on a manually-crafted PROLEX for the target language. The UniSpeech method (Wang et al., 2021b)

---

**Algorithm 1** The JSA algorithm

---

**Input:** Generative model $p_\theta(h, y)$, inference model $q_\phi(h|y)$, and training dataset $\{y_1, \cdots, y_n\}$
  **repeat**
    **Monte Carlo sampling:**
    Draw a random index $\kappa$ over $1, \cdots, n$, pick the data-point $y_\kappa$ along with the cached $\tilde{h}_\kappa$, and use MIS to draw $h_\kappa$;
    **Parameter updating:**
    Update $\theta$ by ascending: $\nabla_\theta \log p_\theta(h_\kappa, y_\kappa)$;
    Update $\phi$ by ascending: $\nabla_\phi \log q_\phi(h_\kappa|y_\kappa)$;
  **until** convergence

---

combines a phoneme-based supervised loss and a self-supervised contrastive loss to improve pre-training, and crosslingual fine-tuning still needs PROLEXs.

**Two-stage ASR.** The two-stage of recognizing speech to phonemes and then to graphemes has been studied for crosslingual ASR (Xue et al., 2023; Lee et al., 2023). The motivation is similar to ours that phoneme units facilitate the learning of shared phonetic representations, making cross-lingual transfer learning effective. However, both studies require a PROLEX for the target language.

**Discrete latent variable models.** Hidden Markov models (HMMs) are classic discrete latent variable models (LVMs) and have been applied to ASR for a long time (Rabiner, 1989). Discrete LVMs are seldom used in recent end-to-end ASR systems, but has been widely used in many other machine learning applications such as dialog systems (Kim et al., 2020; Zhang et al., 2020), program synthesis (Chen et al., 2021), and discrete representation learning (van den Oord et al., 2017).

## 3 BACKGROUND

Consider a latent variable model $p_\theta(h, y)$ for observation $y$ and latent variable $h$, with parameter $\theta$. The joint stochastic approximation (JSA) algorithm (Xu & Ou, 2016; Ou & Song, 2020) is a stochastic extension of the EM algorithm (Dempster et al., 1977) and has demonstrated superior performances particularly in estimating discrete latent variable models. The annoying difficulty of propagating gradients through discrete latent variables is gracefully addressed in JSA.

**Expectation-Maximization (EM) algorithm.** The EM algorithm is an iterative method to find maximum likelihood estimates of parameters for latent variable models. At iteration $t$, the E-step calculates the Q-function $Q(\theta|\theta^{(t-1)}) = E_{p_{\theta^{(t-1)}}(h|y)}[\nabla_\theta log p_\theta(h, y)]$ and the M-step updates $\theta$ by maximizing $Q(\theta|\theta^{(t-1)})$ over $\theta$ or performing gradient ascent over $\theta$ when a closed-form solution is not available. In the E-step, when the expectation in $Q(\theta|\theta^{(t-1)})$ cannot be tractably evaluated, SAEM has been developed (Delyon et al., 1999).

**Stochastic Approximation Version of EM (SAEM).** The SAEM algorithm iterates Monte Carlo sampling and parameter updating. The expectation in the E-step is approximated via Monte Carlo sampling $h' \sim p_{\theta^{(t-1)}}(h|y)$, where $h'$ looks like a stochastic pseudo label for the latent variable. The parameter updating step performs gradient ascent over $\theta$ using $\nabla_\theta log p_\theta(h', y)$, analogous to the M-step in EM.

**MCMC-SAEM.** When exact sampling from $p_{\theta^{(t-1)}}(h|y)$ is difficult, an MCMC-SAEM algorithm has been developed (Kuhn & Lavielle, 2004). MCMC-SAEM draws a sample of the latent $h$ by applying Markov chain Monte Carlo (MCMC) which admits $p_{\theta^{(t-1)}}(h|y)$ as the invariant distribution.

**Joint Stochastic Approximation (JSA).** Given $\theta^{(t-1)}$, the MCMC step in classic MCMC-SAEM is non-adaptive in the sense that the proposal of the transition kernel is fixed. In JSA, an auxiliary amortized inference model $q_\phi(h|y)$ is introduced to approximate the intractable posterior $p_\theta(h|y)$, which is used as the proposal for the MCMC step and adjusted from past realizations of the Markov chain targeting $p_{\theta^{(t-1)}}(h|y)$. So the JSA algorithm amounts to coupling MCMC-SAEM with an adaptive proposal.

The JSA algorithm is summarized in Algorithm 1, which iterates MCMC sampling and parameter updating. In each iteration, we draw a training observation $x_\kappa$ and then sample $h_\kappa$ through Metropo-

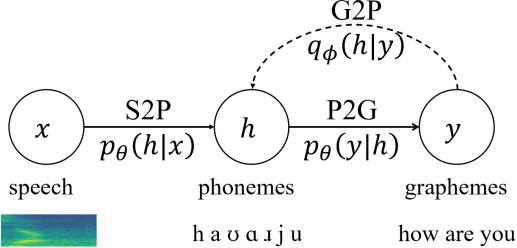

Figure 1: Overview of the latent variable model (SPG), consisting of speech-to-phoneme (S2P) and phoneme-to-grapheme (P2G). Learning SPG without knowing $h$ involves introducing an auxiliary G2P model, denoted by the dashed line.

lis independence sampler (MIS) (Liu, 2001), with $p_\theta(h_\kappa|y_\kappa)$ as the target distribution and $q_\phi(h|y_\kappa)$ as the proposal:

1) Propose $h \sim q_\phi(h|y_\kappa)$;

2) Accept $h_\kappa = h$ with probability $\min\left\{1, \frac{w(h)}{w(\tilde{h}_\kappa)}\right\}$, where

$$w(h) = \frac{p_\theta(h|y_\kappa)}{q_\phi(h|y_\kappa)} \propto \frac{p_\theta(h, y_\kappa)}{q_\phi(h|y_\kappa)}$$

is the usual importance ratio between the target and the proposal distribution and $\tilde{h}_\kappa$ denotes the cached latent state for observation $y_\kappa$.

Algorithm 1 illustrates the simple case that in each iteration, we draw a single data-point and run one step of MIS. In experiments, we apply data minibatching (i.e., drawing a subset of data-points, $x_{\kappa_1}, \cdots, x_{\kappa_m}$) and running MIS with several steps to obtain multiple samples $h_{\kappa_j,1}, \cdots, h_{\kappa_j,T}$ for each data-point $\kappa_j, j = 1, \cdots, m$, where $m$ denotes the minibatch size, and $T$ the sample size. $\{(x_{\kappa_j}, h_{\kappa_j,t})\}$ are pooled for parameter updating.

## 4 METHOD: SPG-JSA

### 4.1 MODEL

Let $(x, y)$ denote the pair of speech and text for an utterance. Specifically, $x$ represents the speech log-mel spectrogram and $y$ the graphemic transcription of $x$. Let $h$ denote the IPA phoneme sequence representing the pronunciation of $x$. In recognizing speech $x$ into text $y$, we treat phonemes $h$ as hidden variables, and construct a latent variable model, which can be decomposed as follows:

$$p_\theta(h, y|x) = p_\theta(h|x)p_\theta(y|h)$$

Basically, the whole model is a conditional generative model from Speech to Phonemes and then to Graphemes, which is referred to as a SPG model. SPG consists a speech-to-phoneme (S2P) model $p_\theta(h|x)$ and a phoneme-to-grapheme (P2G) model $p_\theta(y|h)$.

### 4.2 TRAINING

Training the SPG model from complete data, i.e., knowing $h$, can be easily realized by supervised training. To train S2P and P2G end-to-end (i.e., conducting unsupervised training without knowing $h$), we resort to maximizing the marginal likelihood $p_\theta(y|x)$ and applying the JSA algorithm (Xu & Ou, 2016; Ou & Song, 2020), which has emerged for learning discrete latent variable models with impressive performance.

The original JSA algorithm, described in Algorithm 1, is introduced in an unconditional form, but can be straightforwardly applied in its conditional version, i.e., given $x$. JSA involves introducing an auxiliary inference model to approximate the intractable posterior $p_\theta(h|x, y)$, which, in the ASR task considered in this paper, is assumed to take the form of $q_\phi(h|y)$, i.e., a G2P model.

---

**Algorithm 2** The SPG-JSA algorithm

---

**Input:** S2P model $p_\theta(h|x)$, P2G model $p_\theta(y|h)$, G2P model $q_\phi(h|y)$, training dataset $\{(x, y)\}$
  **repeat**
    Draw a pair of speech and text $(x, y)$;
    $\tilde{h} \leftarrow cache(x, y)$; // get from cache
    **Monte Carlo sampling:**
    Sample $h$ from the proposal $q_\phi(h|y)$;
    Accept $\tilde{h} = h$ with probability $\min\left\{1, \frac{p_\theta(h|x)p_\theta(y|h)}{q_\phi(h|y)} \middle/ \frac{p_\theta(\tilde{h}|x)p_\theta(y|\tilde{h})}{q_\phi(\tilde{h}|y)}\right\}$;
    $cache(x, y) \leftarrow \tilde{h}$; // save to cache
    **Parameter updating:**
    Updating $\theta$ by ascending: $\nabla_\theta[p_\theta(\tilde{h}|x)p_\theta(y|\tilde{h})]$;
    Updating $\phi$ by ascending: $\nabla_\phi q_\phi(\tilde{h}|y)$;
  **until** convergence
  **return** $\theta$ and $\phi$

---

Based on the JSA algorithm (Algorithm 1), we can jointly train the three models (S2P, P2G and G2P), which is summarized in Algorithm 2 (SPG-JSA), where we drop subscript $\kappa$ for simplicity. In each iteration, the stochastic pseudo labels for phonemes are proposed from the G2P model, and got accepted or rejected according to the importance sampling weights:

$$w(h) = \frac{p_\theta(h|x)p_\theta(y|h)}{q_\phi(h|y)} \tag{1}$$

Once we obtain the sampled latent state $h$ from MIS, we can treat them as if being given and calculate the gradients for the S2P, P2G, and G2P models respectively and proceed with parameter updating, similar to the process in supervised training.

**Latent state caching.** In (Ou & Song, 2020), it is found that a two-stage scheme yields fast learning while ensuring convergence. For training with JSA, theoretically we need to cache a latent $h$-sample for each training data-point to maintain a persistent Markov chain. Practically, we run a two-stage scheme. In stage I, we run without caching, i.e. at each iteration, we accept the first proposed sample from $q_\phi(y|x)$ as an initialization and then run MIS with multiple moves. After stage I, we switch to running JSA in its standard manner. In our experiments, we carried out stage I until the model converged, and then switched to stage II to continue training.

**Whistle-JSA.** The SPG-JSA algorithm is general and is in fact an unsupervised learning over $(x, y)$ without the need to know $h$. It is challenging to run this purely unsupervised form from scratch in the ASR task considered in this paper, which involves very high-dimensional latent space. Two additional techniques are incorporated to add inductive bias into model training. First, the S2P model is initialized from a pre-trained phoneme-based multilingual S2P backbone, called Whistle (Yusuyin et al., 2024), which have been shown to have good phoneme classification ability. In our experiments, the S2P, P2G, and G2P models are all implemented by CTC (Graves et al., 2006), which will be more detailed in Section 5.2. Second, we assume that 10 minutes of transcribed speech with phoneme labels are available, which takes much less labor than compiling a complete PROLEX for a target language. Thus, we combine supervised learning over 10 minutes speech with phoneme labels and unsupervised learning over a much larger dataset without phoneme labels. Bootstrapping from a good S2P backbone (Whistle) and providing few-shots samples of latent variables (10 minutes of phoneme labels) is found to be important to make Whistle-JSA, as a specific instantiation of the general SPG-JSA approach, successfully work in the challenging task of crosslingual ASR.

### 4.3 DECODING

In testing, the S2P model first decodes out the phoneme sequence $h$ using BeamSearch and selects the best beam as input for the P2G model. Then, the P2G model also employs BeamSearch to decode the speech recognition results, which is named as "w/o LM" result. Similar to the subword-based Whistle model (Yusuyin et al., 2024), we use an n-gram language model for WFST-based decoding, which is named as "w LM" result.

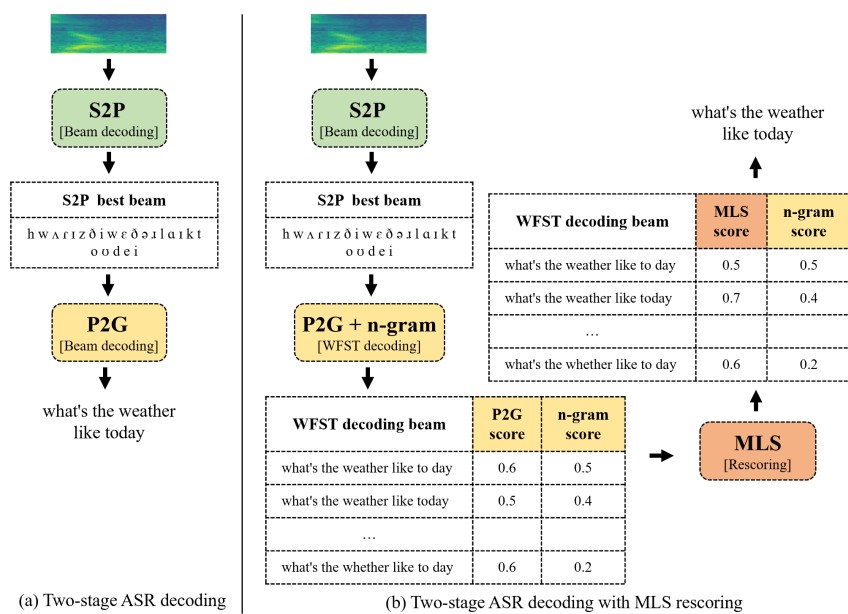

Figure 2: Illustration of decoding in Whistle-JSA. (a) Decoding without LM and (b) MLS rescoring.

**Marginal likelihood scoring.** Note that the training objective of the JSA algorithm is maximizing the marginal likelihood $p_\theta(y|x)$. The decoding procedure in Section 4.3 is a crude approximation to the training objective, which is referred to as "crude decoding". So we propose a new decoding algorithm, called "decoding with marginal likelihood scoring" (MLS). It consists of the following steps: 1) S2P takes in the audio $x$ and outputs the BeamSearch best result $\hat{h}$; 2) P2G takes in the $\hat{h}$ and generates an n-best list of candidates $\hat{y}$ using WFST decoding; 3) G2P takes in each candidate hypothesis $\hat{y}$ and propose $l$ samples $h$ from $q_\phi(\hat{h}|\hat{y})$; 4) The marginal likelihood can be estimated with importance weights (Xu & Ou, 2016), as shown in Eq. 1; 5) Each candidate hypothesis $\hat{y}$ is rescored using a sum of the estimated marginal likelihood and the weighted LM score. In summary, the above steps can be written as:

$$y^* = \arg\max_{\hat{y}} \log \sum_{i=0}^{l} \frac{p_\theta(h_i|x)p_\theta(\hat{y}|h_i)}{q_\phi(h_i|\hat{y})} + \lambda \log P_{\text{LM}}(\hat{y}) \qquad (2)$$

where $\hat{y}$ takes from the n-best list from crude decoding, and $\lambda$ is LM weight. Additionally, note that crude decoding only uses the single best S2P result to fed to P2G for decoding, which is easily prone to error propagation. Decoding with MLS overcomes this drawback by scoring with multiple $h$.

**Improving P2G via data augmentation.** Note that during the whistle-JSA training, as the models gradually converge, the diversity of phoneme sequences sampled by MIS decreases. The P2G model is gradually trained with less noisy input, compared with the input fed to P2G in testing. In order to improve the robustness of the P2G model, we further augment the P2G model after the Whistle-JSA training. Particularly, we decode 128 best phoneme sequences by S2P BeamSearch decoding and pair them with text labels, which serve as augmented data to further train the P2G model.

## 4.4 LANGUAGE DOMAIN ADAPTATION

Note that after Whistle-JSA training, we can use the auxiliary G2P model to generate phoneme labels on pure text. Below, we take the language domain adaptation task as an example to introduce the bonus brought by the G2P model.

Text-only data is easier to obtain than transcribed speech data. In cross-domain ASR, a common approach is to train external language models for language domain adaptation. In contrast, in Whistle-JSA, we can use the G2P model to generate 64 best phoneme labels through BeamSearch decoding, and then use the pairs of phonemes and text to continue adapting the P2G model. Then, we use

the original S2P, the adapted P2G, and the cross-domain language model for speech recognition on cross-domain audio, which is found to outperform the standard practice of only doing language model fusion.

# 5 EXPERIMENT

## 5.1 DATASETS

**Common Voice** (Ardila et al., 2020) is a large multilingual speech corpus, with spoken content taken primarily from Wikipedia articles. We conduct experiments on the Common Voice dataset released at September 2022 (v11.0). We select Polish (pl) and Indonesian (id) for Whistle-JSA experiments, which were not used in Whistle pre-training. Polish has 130 hours of training data, while Indonesian has 20 hours, with an average sentence length of 4.3 and 4.5 seconds, respectively. We selected 100 text sentences from the training set of each language and converted them into phonetic annotations using a publicly available phonemizer (Novak et al., 2016). In the Whistle-JSA experiment, we utilized all the audio data from the two language training sets along with the corresponding text transcriptions and 100 sentences (about 10 minutes) of phonetic labels.

**VoxPopuli** (Wang et al., 2021a) is a multilingual speech dataset of parliamentary speeches in 23 European languages from the European Parliament. The Polish training set consists of 94.5 hours (or 710,000 words) transcribed speech data, with an average sentence length of 10 seconds. We use the training set texts for language domain adaptation experiments. Additionally, the Polish validation set is used for model selection, and the test set is used for evaluation.

**Indonesian in-house data.** We conducted Indonesian language domain adaptation experiments using our in-house dataset (VoxPopuli does not include Indonesian). This dataset consists of 798 hours (or 6.16 M words) transcribed speech data, with an average sentence length of 5.18 seconds. We use the training set texts for language domain adaptation experiments. Additionally, the validation set is used for model selection, and the test set is used for evaluate the experimental results.

## 5.2 SETUP

For phoneme-based models, both of the polish and Indonesian alphabet size of phonemes is 35. For subword-based models, both of the polish and Indonesian alphabet size of subwords is 500. All text normalization and phonemization strategies are consistent with the Whistle work (Yusuyin et al., 2024). For each language, we use its transcripts to separately train a word-level n-gram language model for WFST-based decoding.

In the experiments, the S2P, P2G, and G2P models are all based on CTC. The Whistle-small 90M pre-trained model [1] is used to initialize the S2P model. Both the G2P and P2G models use 8-layer Transformer encoders with dimension 512. We set the self-attention layer to have 4 heads with 512-dimension hidden states, and the feed-forward network (FFN) dimension to 1024. All experiment are taken with the CAT toolkit (An et al., 2020). The learning rate for Whistle-JSA is set to 3e-5, and when the validation loss does not decrease 10 epochs, the learning rate is multiplied by 0.5, training stop until it reaches 1e-6. We extract 80-dimension FBank features from audio (resampled to 16KHz) as inputs to the S2P model. A beam size of 16 is used for S2P and P2G decoding in testing. We average the three best-performing checkpoints on the validation set for testing.

# 6 RESULT AND ANALYSIS

## 6.1 WHISTLE-JSA RESULTS

Baseline results are taken from (Yusuyin et al., 2024), including monolingual phoneme-based training and subword-based training. The phoneme-based training utilized full phonetic annotations for 130 hours of Polish and 20 hours of Indonesian data. The phoneme-based Whistle-small pre-trained

---

[1] https://github.com/thu-spmi/CAT/tree/master/egs/cv-lang10/exp/
Multilingual/Multi._phoneme_S

Table 1: PERs (%) and WERs (%) for Whistle-JSA experiment on Common Voice dataset. FT: fine-tuning. MLS: marginal likelihood scoring. [†] denotes results from (Yusuyin et al., 2024).

| Exp. | Polish | | | | Indonesian | | | |
|---|---|---|---|---|---|---|---|---|
| | PER | w/o LM | w LM | *MLS* | PER | w/o LM | w LM | *MLS* |
| Mono. phoneme FT [†] | 2.82 | - | 4.97 | - | 5.74 | - | 3.28 | - |
| Mono. subword FT [†] | - | 19.38 | 7.12 | - | - | 31.96 | 10.85 | - |
| Whistle phoneme FT [†] | 1.97 | - | 4.30 | - | 4.79 | - | 2.43 | - |
| Whistle subword FT [†] | - | 5.84 | 3.82 | - | - | 12.48 | 2.92 | - |
| Whistle-JSA | 17.58 | 15.70 | 5.66 | 4.51 | 20.55 | 17.15 | 4.68 | 3.34 |
| + continue with cache | 17.39 | 14.07 | 5.49 | 4.23 | 20.66 | 16.31 | 4.58 | 3.33 |
| + P2G augmentation | 17.39 | 7.23 | 5.03 | 3.95 | 20.66 | 9.68 | 3.95 | 3.04 |

model were further fine-tuned with either phoneme labels or subword labels for crosslingual speech recognition. Phoneme fine-tuning used full phonetic annotations.

In the following, we introduce the Whistle-JSA crosslingual ASR experiments with only 10 minutes of data per language having phoneme annotations. The Whistle-JSA experiments were divided into three settings. The first setting is Whistle-JSA training without caching. The second setting is continued training with caching. In the third setting, we further improve P2G via data augmentation, as described in Section 4.3. In the beginning of the Whistle-JSA training, we first fine-tuned the Whistle model on 10 minutes of phoneme labels to initialize the S2P model. Subsequently, this S2P model was utilized to generate phoneme pseudo-labels on the training set, which were then used to train the P2G and G2P models for initialization.

For the Polish results shown in Table 1, we can see that after MLS decoding, Whistle-JSA training achieves performance surpassing two monolingual models and approaching the results of crosslingual phoneme fine-tuning. Upon continue training with cache, it can exceed crosslingual phoneme fine-tuning which needs 130 hour phoneme labels. The result by P2G augmentation closely matches that of crosslingual subword fine-tuning. According p-value (0.5768) by matched-pair significance test (Gillick & Cox, 1989), there is no statistically significant difference between Whistle subword fine-tuning result and Whistle-JSA result (3.82 vs 3.95).

For Indonesian results in Table 1, similar to Polish, Whistle-JSA achieved results close to those of crosslingual subword fine-tuning, and according to p-value (0.1656), there is no statistically significant difference between Whistle-JSA and Whistle subword fine-tuning (2.92 vs 3.04).

Remarkably, in Indonesian, Whistle phoneme-based fine-tuning demonstrates better performance than Whistle subword fine-tuning. As analyzed in (Yusuyin et al., 2024), when crosslingual fine-tuning data is more limited (Indonesian has 20 hours of data vs Polish 130 hours), phoneme-based fine-tuning is more data-efficient and performs better than subword fine-tuning.

Experiments conducted on two languages from different language families and with varying amounts of data indicate that the two-stage ASR network from S2P to P2G proposed in this paper, using only 10 minutes of phoneme annotations, achieves competitive performance with single-stage crosslingual subword fine-tuning. This demonstrates the superiority and versatility of Whistle-JSA.

## 6.2 Language Domian Adaptation Results

As shown in Table 2, for Polish, we test our models on VoxPopuli Polish test set, while both Whistle and Whistle-JSA models is train on the Common Voice dataset. The CommonVoice dataset is comprised of texts from Wikipedia, recorded by users on mobile devices, while the VoxPopuli dataset consists of audio recordings of speeches from the European Parliament. Notably, 61.5% of the words in the VoxPopuli Polish training set do not appear in the CommonVoice vocabulary list, and 31.5% of the words in the test set are also absent. This indicates significant differences between the

Table 2: WERs (%) of language domain adaptation (LDA) experiments on cross-domain dataset. The FT denotes fine-tuning. The MLS denotes marginal likelihood scoring.

| Exp. | Polish | | | Indonesian | | |
|---|---|---|---|---|---|---|
| | w/o LM | w LM | *MLS* | w/o LM | w LM | *MLS* |
| Whistle subword FT on CV | 33.46 | 22.58 | - | 43.69 | 12.39 | - |
| Whistle-JSA on CV | 31.59 | 24.93 | 22.19 | 43.27 | 16.76 | 13.86 |
| + LDA training | 25.23 | 21.66 | 20.36 | 37.78 | 13.83 | 12.14 |

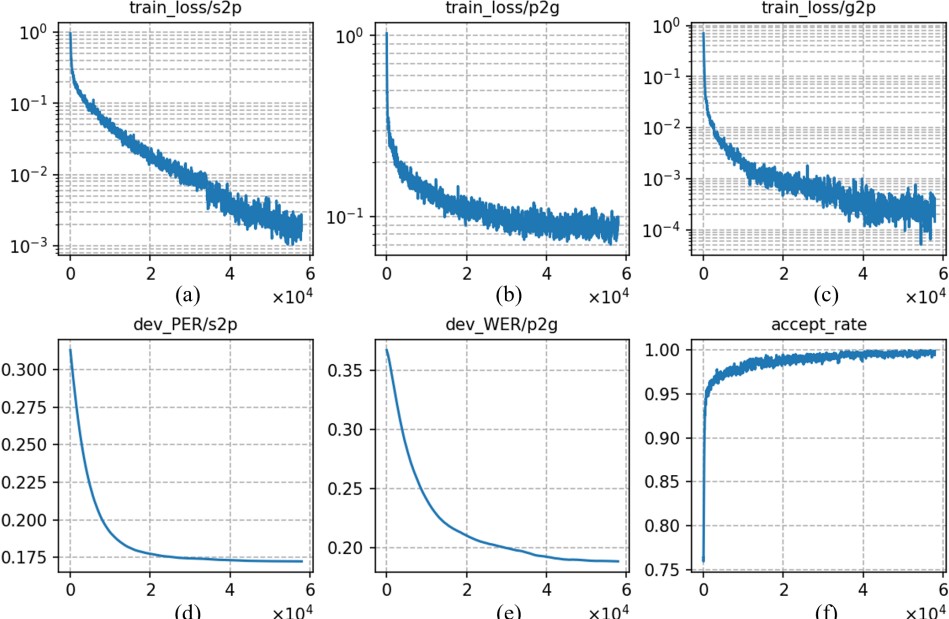

Figure 3: Plots of training and validation curves in Whistle-JSA training on Common Voice polish data. (a), (b), (c) represent the train losses of the S2P, P2G, and G2P models in the Whistle-JSA training, respectively. (d) and (e) are the error rates of S2P and P2G models in the validation set. (f) represents the ratio of the number of samples accepted by the MIS sampler to the total number of samples proposed by G2P in one iteration.

two datasets in terms of linguistic context, vocabulary, recording equipment, and average sentence length. We only use the text from the VoxPopuli training set and train a word-level 4-gram language model for language model fusion.

The first row at Table 2 shows the results of testing Whistle subword fine-tuning model directly with cross domain language model integration, which is a common method used in cross-domain speech recognition. We then test Whistle-JSA model directly without further training in the second row of Table 2. Comparing the two result on Polish reveals that the Whistle-JSA model performs better on cross-domain ASR tasks, indicating its stronger robustness. We further apply the domain adaptation method, introduced in Section 4.4, to continue training the P2G model on VoxPopuli training text. The result clearly demonstrates the advantage of Whistle-JSA, and its performance far exceeds that of traditional language domain adaptation method by 9.8% error rate reduction (22.58 vs 20.36).

For Indonesian, our in-house Indonesian dataset is from audio books, which has a clear domain difference from the CommonVoice dataset. We continued training the P2G model on in-house text-only data using the Whitsle-JSA model, and it also outperformed the Whistle sub-word fine-tuning result, though with smaller improvement compared with the Polish experiment.

Table 3: Performance comparison of different sample sizes in Polish unsupervised Whistle-JSA training.

| n-samples | PER | WER | |
|---|---|---|---|
| | | w/o LM | w LM |
| 10 | 17.58 | 15.70 | 5.66 |
| 50 | 17.14 | 13.57 | 5.58 |
| 100 | 17.03 | 13.21 | 5.46 |

### 6.3 Analysis and Ablation

To provide a more intuitive description of the Whistle-JSA training process, Figure 3 shows the changes in several key indicators over the number of training steps. It can be seen that the training loss of all three models and validation error rates gradually decrease when using the method of Algorithm 2 for parameter iteration. Through Whistle-JSA training, compared to the model fine-tuned with only 10 minutes of phonetic labels, which is the initial model in this experiment, the Whistle-JSA model achieves a relative PER reduction of 45% and a WER reduction of 48% on the validation set. As the three models gradually converge, the diversity of samples that sampled out of the G2P begins to decrease, and the proportion of phoneme sequences accepted by the MIS sampler converges to 100% as shown in Figure 3 (f).

Table 3 shows ablation experiments with different MIS sample sizes. As the number of samples increases, both PER and WER of the model significantly decrease. Compared to 10 samples, at 50 samples, PER decreases by 2.5% and WER by 1.4%; at 100 samples, PER decreases by 3.1% and WER by 3.5%. It is worth noting that increasing the number of samples provides more diverse sampling, which benefits random approximation models in searching a wider latent space, but a balance needs to be found with computational cost. All Whistle-JSA experiments in this paper use a sample size of 10, indicating that there is still have the potential for further improvement.

## 7 Conclusion and Future Work

In this paper, we achieve crosslingual speech recognition based on phonemes without a pronunciation lexicon. By treating phonemes as discrete latent variables, S2P model and P2G model as latent variable models, and introducing CTC based G2P model as an auxiliary inference model, we utilize the JSA algorithm to jointly train these three networks. We called this approach as Whistle-JSA. This paper also proposes a MLS decoding approach that rescores each candidate hypothesis using the marginal likelihood score and language model score. We also propose a P2G augmentation strategy that uses the n-best results decoded by S2P to improve the robustness of the P2G model towards input data. In crosslingual experiments with two languages, using only 10 minutes of data, the Whistle-JSA method achieved results close to full data (130 and 20 hours) phonetic supervision, and perform comparable to crosslingual subword fine-tuning. We further conduct language domain adaptation experiments. The results show that the Whistle-JSA model outperform the standard language model fusion approach via the auxiliary support of the G2P model. In the future, we plan to train phoneme-based pre-trained models using the Whistle-JSA method on more languages.

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
