# OpenReview forum: "Pronunciation-Lexicon Free Training for Phoneme-based Crosslingual ASR via Joint Stochastic Approximation"
_ICLR.cc/2025/Conference — ICLR 2025 Conference Withdrawn Submission_

### Official Review · Reviewer_ZQTR · 2024-11-02

**Soundness:** 3
**Presentation:** 3
**Contribution:** 3
**Rating:** 1
**Confidence:** 4

**Summary:**

This work presents a cross-lingual phoneme-based asr model by considering phonemes as latent variables. They apply an approach called
 joint stochastic approximation to estimate latent phonemes. The formulation is well-motivated and I think this direction is worth investigating. However, many paragraphs in this work is copied paste from another paper, so I am not able to trust the remaining sections and cannot provide further review on it

**Strengths:**

I am not able to fully access the paper due to the copy-paste issue mentioned in the weakness section below

**Weaknesses:**

Many paragraphs from Background is copied paste from the following paper (which authors cited as well though)
https://arxiv.org/pdf/2005.14001

For example, Expectation-Maximization (EM) algorithm subsection in this paper is identical to the first parargraph on the top-right 4th page in the previously mentioned papers. The other paragraphs in the Background section also copied paste non-trivial numbers of sentences from this papers as well.

Authors should rewrite them in their own words if they think those contents are important. It is not ethical and dangerous to copy from others. The copied paragraph even copied wrong descriptions already exist in the original paper. Both the original source and copied paragraph contain wrong formulation about Q-function: it is formulated as $Q=E[\nabla \log p]$ in both paragraphs, however, the correct Q-function does not contain derivative in its expectation

**Questions:**

No questions

**Details Of Ethics Concerns:**

See the weakness section above

---

### Official Review · Reviewer_NTF8 · 2024-11-02

**Soundness:** 3
**Presentation:** 3
**Contribution:** 3
**Rating:** 6
**Confidence:** 4

**Summary:**

This paper focuses on cross-lingual ASR without using a phoneme lexicon. In this work, a challenging setting is considered - it is assumed that the training data consists of speech and grapheme text pairs, and a very limited amount of transcribed phonemes are available (can be as small as 10 mins).

Phonemes are modeled as latent variables, and the entire ASR system is factorized into a S2P model, a P2G model and a G2P model (as an auxiliary inference model). An algorithm called Whistle-JSA, which is based on Joint Stochastic Approximation, is proposed to train the system. Several training stages are involved, and a marginal likelihood scoring decoding is proposed to improve the performance.  The empirical results show that, trained with 10-min labeled phonemes, in terms of WER, it performs close to fine-tuning with full phoneme annotations, and on par with fine-tuning with full subword supervision.

**Strengths:**

1. The SPG-JSA algorithm to train S2P, P2G and G2P models is novel.
2. Experiments are solid.
3. The paper is well-written. Experimental setups are detailed and clear.

**Weaknesses:**

1. I think the main weakness is that the algorithm is rather complicated. First, it requires some pre-trained models to initialize the S2P, P2G and G2P models separately. As the authors stated “we first fine-tuned the Whistle model on 10 minutes of phoneme labels to initialize the S2P model. Subsequently, this S2P model was utilized to generate phoneme pseudo-labels on the training set, which were then used to train the P2G and G2P models for initialization.” Second, training the entire model requires 2 stages, as described in Section 4.2. Third, due to the factorization of S2P, P2G and G2P, the MLS decoding is also complicated. I would expect a quite slow inference due to separate beam searches and re-scoring. This brings me to my first question below.
2. From the results in Table 1, the proposed method does not outperform the simple subword fine-tuning (which also only assumes (speech, text) pairs available, if i understand correctly.

**Questions:**

1. How does it compare to a much simpler pseudo-labeling approach? After obtaining the initialization of S2P, P2G and G2P via fine-tuning and pseudo-labeling, I could imagine that the G2P model (instead of the S2P model) can be used to generate phoneme pseudo labels, which can then be used to train S2P and P2G models.
2. The performance of the resulting G2P model’s performance is not shown. How does it compare to the G2P initialization performance?
3. In general, I’m a bit confused about the necessity of producing phonemes for ASR. Nowadays, the mainstream approach is end-to-end ASR. This work assumes available (speech, text) pairs. In this case, a simple end-to-end fine-tuning (e.g. SSL models) with text targets can be easily done. Is it worse than considering phonemes? Also, as the results show, using subwords for ASR fine-tuning already slightly outperforms Whistle-JSA (Table 1) in terms of WER. Assuming texts available, one can train even better subword tokenizers. Further, one can even use characters as targets without any tokenization. If the end goal is ASR, what is the motivation/benefit of explicitly modeling phonemes? For such a complicated algorithm, I think more justification is needed to convince the audience.

---

### Official Review · Reviewer_ryBQ · 2024-11-04

**Soundness:** 3
**Presentation:** 3
**Contribution:** 3
**Rating:** 5
**Confidence:** 4

**Summary:**

This paper proposes Whistle-JSA, which aims to train an ASR model with minimal dependency on a pronunciation lexicon. The method views the phonetic representation of an utterance as hidden variables and uses a joint-stochastic approximation method to jointly train S2P, P2G, and G2P components. They propose methods to utilize beam-search as well as multiple hypotheses, as well as data augmentation techniques to improve results. They show that they're able to get comparable results with 10 minutes of phoneme supervision to existing methods that use >100 hours of data.

**Strengths:**

The paper is mostly well-written and easy to understand. The technique is described well with the necessary details for reproduction. The theory behind the paper isn't new, though the authors carefully design a set of algorithms for the particular ASR problem, making the novelty OK. The results seem quite impressive considering the amount of the phoneme-labeled data was drastically reduced with the proposed method.

**Weaknesses:**

The evaluation was done on two languages, and only with CTC model, which isn't enough to convince me fully of the usefulness of the method.

 And, given the set of experiments reported, I'm not fully convinced that JSA + beam-search + augmentation + multiple-hypotheses reranking is really necessary.

I understand that initially, due to the small amount of phoneme-labeled data, training a separate S2P model might be necessary to bootstrap something useful. And it seems those search techniques and reranking methods all are proposed to solve the issue that S2P might give bad phoneme predictions which can propage to grapheme level. The essence of this limitation is that, once you perform a search using the S2P model, the problem is descretized and gradients can't propagate across those components, and therefore there was the need to include diverse hypotheses so that eventually the correct grapheme can be generated.

I feel an important alternative method that this method needs to compare with is, after a simple bootstrapping, instead of performing a search on the S2P model, we feed the last layer output (prior to the last affine before softmax) to the P2G model, and let the model train in an end-to-end way. We can either detach the gradient between the two components or maybe still let the S2P component train, possibly with a smaller learning rate. This model would be much simpler than what is proposed in this paper, and more efficient to run. If this simpler model works as well, then I'm not fully convinced the model proposed in this paper should be preferred since it's much more complex.

Some typos: Domian -> Domain. Whitsle (appeared once) should be whistle.

**Questions:**

Please see the weakness section.

---

### Official Review · Reviewer_Yawf · 2024-11-04

**Soundness:** 2
**Presentation:** 3
**Contribution:** 2
**Rating:** 3
**Confidence:** 3

**Summary:**

The Speech2Phone (S2P) model (e.g., Whistle) can convert arbitrary speech into phonetic representations. This paper further introduces a phoneme-to-grapheme (P2G) module to convert phonemes into specific languages. These processes enable recognition without requiring phonetic tagging for each language. The authors also introduce a grapheme-to-phoneme (G2P) model to improve latent-variable learning. They apply a joint stochastic approximation (JSA) method to simultaneously update the S2P, G2P, and P2G models. Experiments are conducted on two languages, Polish and Indonesian.

**Strengths:**

The proposed SPG-JSA method efficiently achieves cross-lingual transfer.

**Weaknesses:**

- The paper addresses the multilingual speech recognition problem but does not compare its approach to advanced models like Whisper and Seamless, either in terms of speed or performance.
- The contribution is limited, as the backbone of the work is the S2P model (Whistle), with JSA simply combining existing components.
- While the paper focuses on multilingual capabilities, the experiments are conducted only on Polish and Indonesian.

**Questions:**

How would the performance differ if Whisper or Seamless were used instead?

---

### Note · Authors · 2024-11-23

I have read and agree with the venue's withdrawal policy on behalf of myself and my co-authors.